# Learning bounded degree polytrees with samples

## Abstract

We establish finite-sample guarantees for efficient proper learning of bounded-degree *polytrees*, a rich class of high-dimensional probability distributions and a subclass of Bayesian networks, a widely-studied type of graphical models. Very recently, Bhattacharyya et al. [2021] obtained finite-sample guarantees for recovering tree-structured Bayesian networks, i.e., 1-polytrees. We considerably extend their results by providing an efficient algorithm which learns $d$-polytrees in polynomial time and sample complexity when the in-degree $d$ is constant, provided that the underlying undirected graph (skeleton) is known. We complement our algorithm with an information-theoretic lower bound, showing that the dependence of our sample complexity is nearly tight in both the dimension and target accuracy parameters.

## 1 Introduction

Distribution learning, or density estimation, is the task of obtaining a good estimate of some unknown underlying probability distribution $P$ from observational samples. Understanding which classes of distributions could be or could not be learnt efficiently is a fundamental problem in both computer science and statistics, where efficiency is measured in terms of *sample* (data) and *computational* (time) complexities.

*Bayesian networks* (or *Bayes nets* in short) represent a class of high-dimensional distributions that can be explicitly described by how each variable is be generated sequentially in a directed fashion. Being interpretable, Bayes nets have been used to model beliefs in a wide variety of domains (e.g. see [Jensen and Nielsen, 2007, Koller and Friedman, 2009] and references therein). A fundamental problem in computational learning theory is to identify families of Bayes nets which can be learned efficiently from observational data.

Formally, a Bayes net is a probability distribution $P$, defined over some directed acyclic graphs (DAG) $G = (V, E)$ on $|V| = n$ nodes that factorizes according to $G$ (i.e. Markov with respect to $G$) in the following sense: $P(v_1, \ldots, v_n) = \prod_{v_1, \ldots, v_n} P(v \mid \pi(v))$, where $\pi(v) \subseteq V$ are the parents of $v$ in $G$. While it is well-known that given the DAG (structure) of a Bayes net, there exists sample-efficient algorithms that output good hypotheses [Dasgupta, 1997, Bhattacharyya et al., 2020], there is no known computationally efficient algorithms for obtaining the DAG of a Bayes net. In fact, it has long been understood that Bayes net structure learning is computationally expensive, in various general settings [Chickering et al., 2004]. However, these hardness results fall short when the goal is learning the distribution $P$ in the probabilistically approximately correct (PAC) [Valiant, 1984] sense (with respect to, say, KL divergence or total variation distance), rather than trying to recover an exact graph from the Markov equivalence class of $P$.

*Polytrees* are a subclass of Bayesian networks where the undirected graph underlying the DAG is a forest, i.e., there is no cycle for the undirected version of the DAG; a polytree with maximum in-degree $d$ is also known as a $d$-polytree. With an infinite number of samples, one can recover the

DAG of a non-degenerate polytree in the equivalence class with the Chow–Liu algorithm [Chow and Liu, 1968] and some additional conditional independence tests [Rebane and Pearl, 1988]. However, this algorithm does *not* work in the finite sample regime. The only known result for learning polytrees with finite sample guarantees is for 1-polytrees [Bhattacharyya et al., 2021]. Furthermore, in the agnostic setting, when the goal is to find the closest polytree distribution to an arbitrary distribution $P$, the learning problem becomes NP-hard [Dasgupta, 1999].

In this work, we investigate what happens when the given distribution is a $d$-polytree, for $d > 1$. *Are d-polytrees computationally hard to learn in the realizable PAC-learning setting?* One motivation for studying polytrees is due to a recent work of Gao and Aragam [2021] which shows that polytrees are easier to learn than general Bayes nets due to the underlying graph being a tree, allowing typical causal assumptions such as faithfulness to be dropped when designing efficient learning algorithms.

**Contributions.** Our main contribution is a sample-efficient algorithm for proper Bayes net learning in the realizable setting, when provided with the ground truth skeleton (i.e., the underlying forest). Crucially, our result does not require any distributional assumptions such as strong faithfulness, etc.

**Theorem 1.** *There exists an algorithm which, given $m$ samples from a polytree $P$ over $\Sigma^n$, accuracy parameter $\varepsilon > 0$, failure probability $\delta$, as well as its maximum in-degree $d$ and the explicit description of the ground truth skeleton of $P$, outputs a $d$-polytree $\hat{P}$ such that $d_{\mathrm{KL}}(P \parallel \hat{P}) \leq \varepsilon$ with success probability at least $1 - \delta$, as long as*

$$m = \tilde{\Omega}\left( \frac{n \cdot |\Sigma|^{d+1}}{\varepsilon} \log \frac{1}{\delta} \right) \ .$$

*Moreover, the algorithm runs in time polynomial in $m$, $|\Sigma|^d$, and $n^d$.*

We remark that our result holds when even given only an upper bound on the true in-degree $d$. In particular, our result provides, for constant $|\Sigma|$, $d$, an upper bound of $\tilde{O}(n/\varepsilon)$ on the sample complexity of learning $O(1)$-polytrees. Note that this dependence on the dimension $n$ and the accuracy parameter $\varepsilon$ are optimal, up to logarithmic factors: indeed, we establish in Theorem 15 an $\Omega(n/\varepsilon)$ sample complexity lower bound for this question, even for $d = 2$ and $|\Sigma| = 2$.[1]

We also state sufficient conditions on the distribution that enable recovery of the ground truth skeleton. Informally, we require that the data processing inequality hold in a strong sense with respect to the edges in the skeleton graph. Under these conditions, combining with our main result in Theorem 1, we obtain a polynomial-time PAC algorithm to learn bounded-degree polytrees from samples.

**Other related work.** Structure learning of Bayesian networks is an old problem in machine learning and statistics that has been intensively studied; see, for example, Chapter 18 of Koller and Friedman [2009]. Many of the early approaches required faithfulness, a condition which permits learning of the Markov equivalence class, e.g. Spirtes and Glymour [1991], Chickering [2002], Friedman et al. [2013]. Finite sample complexity of such algorithms assuming faithfulness-like conditions has also been studied, e.g. Friedman and Yakhini [1996]. An alternate line of more modern work has considered various other distributional assumptions that permits for efficient learning, e.g., Chickering and Meek [2002], Hoyer et al. [2008], Shimizu et al. [2006], Peters and Bühlmann [2014], Ghoshal and Honorio [2017], Park and Raskutti [2017], Aragam et al. [2019], with the latter three also showing analyzing finite sample complexity. Specifically for polytrees, Rebane and Pearl [1988], Geiger et al. [1990] studied recovery of the DAG for polytrees under the infinite sample regime. Gao and Aragam [2021] studied the more general problem of learning Bayes nets, and their sufficient conditions simplified in the setting of polytrees. Their approach emphasize more on the exact recovery, and thus the sample complexity has to depend on the minimum gap of some key mutual information terms. In contrast, we allow the algorithm to make mistakes when certain mutual information terms are too small to detect for the given sample complexity budget and achieve a PAC-type guarantee. As such, once the underlying skeleton is discovered, our sample complexity only depends on the $d, n, \varepsilon$ and not on any distributional parameters.

There are existing works on Bayes net learning with tight bounds in total variation distance, with a focus on sample complexity (and not necessarily computational efficiency); for instance, [Canonne et al., 2020]. Acharya et al. [2018] consider the problem of learning (in TV distance) a bounded-degree causal Bayesian network from interventions, assuming the underlying DAG is known.

---

[1]We remark that [Bhattacharyya et al., 2021, Theorem 7.6] implies an $\Omega(\frac{n}{\varepsilon} \log \frac{n}{\varepsilon})$ sample complexity lower bound for the analogous question when the skeleton is unknown and $d = 1$.

 **Outline of paper.** We begin with some preliminary notions and related work in Section 2. Section 3
then shows how to recover a polytree close in KL divergence, assuming knowledge of the skeleton
and maximum in-degree. Section 4 gives sufficient conditions to recover the underlying skeleton from
samples, while Section 5 provides our sample complexity lower bound. We conclude in Section 6
with some open directions and defer some full proofs to the appendix.

## 2 Preliminaries and tools from previous work

### 2.1 Preliminary notions and notation

We write the disjoint union as $\dot{\cup}$. For any set $A$, let $|A|$ denotes its size. We use hats to denote
estimated quantities, e.g., $\hat{I}(X;Y)$ will be the estimated mutual information of $I(X;Y)$. We employ
the standard asymptotic notation $O(\cdot)$, $\Omega(\cdot)$ $\Theta(\cdot)$, and write $\tilde{O}(\cdot)$ to omit polylogarithmic factors.
Throughout, we identify probability distributions over discrete sets with their probability mass
functions (pmf). We use $d^*$ to denote the true maximum in-degree of the underlying polytree.

### 2.2 Probability distribution definitions

We begin by defining KL divergence and squared Hellinger distances for a pair of discrete distributions
with the same support.

**Definition 2** (KL divergence and squared Hellinger distance). For distributions $P, Q$ defined on
the same discrete support $\mathcal{X}$, their KL divergence and squared Hellinger distances are defined as
$d_{\mathrm{KL}}(P \parallel Q) = \sum_{x \in \mathcal{X}} P(x) \log \frac{P(x)}{Q(x)}$ and $d_{\mathrm{H}}^2(P, Q) = 1 - \sum_{x \in \mathcal{X}} \sqrt{P(x) \cdot Q(x)}$ respectively.

Abusing notation, for a distribution $P$ on variables $X = \{X_1, \ldots, X_n\}$, we write $P_S$ to mean the
projection of $P$ to the subset of variables $S \subseteq X$ and $P_G$ to mean the projection of $P$ onto a graph $G$.
More specifically, we have $P_G(x_1, \ldots, x_n) = \prod_{x \in X} P(x \mid \pi_G(x))$ where $\pi_G(x)$ are the parents of $x$
in $G$. Note that $P_G$ is the closest distribution[2] on $G$ to $P$ in $d_{\mathrm{KL}}$, i.e. $P_G = \mathrm{argmin}_{Q \in G} d_{\mathrm{KL}}(P \parallel Q)$.
By Chow and Liu [1968], we also know that

$$d_{\mathrm{KL}}(P, P_G) = -\sum_{i=1}^n I(X_i; \pi_G(X_i)) - H(P_X) + \sum_{i=1}^n H(P_{X_i}) \,, \tag{1}$$

where $H$ is the entropy function. Note that only the first term depends on the graph structure of $G$.

By maximizing the sum of mutual information (the negation of the first term in (1)), we can obtain an
$\varepsilon$-approximated graph $G$ such that $d_{\mathrm{KL}}(P \parallel P_G) \leq \varepsilon$. In the case of tree-structured distributions, this
can be efficiently solved by using any maximum spanning tree algorithm; a natural generalization to
bounded degree bayes nets remains open due to the hardness of solving the underlying optimization
problem [Höffgen, 1993]. If any valid topological ordering of the target Bayes net $P$ is present, then
an efficient greedy approach is able to solve the problem.

**Definition 3** ((Conditional) Mutual Information). Given a distribution $P$, the mutual information of
two random variables $X$ and $Y$, supported on $\mathcal{X}$ and $\mathcal{Y}$ respectively, is defined as

$$I(X;Y) = \sum_{x \in \mathcal{X}, y \in \mathcal{Y}} P(x,y) \cdot \log \left( \frac{P(x,y)}{P(x) \cdot P(y)} \right) \,.$$

Conditioning on a third random variable $Z$, supported on $\mathcal{Z}$, the conditional mutual information is
defined as:

$$I(X;Y \mid Z) = \sum_{x \in \mathcal{X}, y \in \mathcal{Y}, z \in \mathcal{Z}} P(x,y,z) \cdot \log \left( \frac{P(x,y,z) \cdot P(z)}{P(x,z) \cdot P(y,z)} \right) \,.$$

By adapting a known testing result from [Bhattacharyya et al., 2021, Theorem 1.3], we can obtain the
following corollary, which we will use. We provide the full derivation in the supplementary materials.

---

[2]One can verify this using Bhattacharyya et al. [2021, Lemma 3.3]: For any distribution $Q$ defined on graph
$G$, we have $d_{\mathrm{KL}}(P \parallel Q) - d_{\mathrm{KL}}(P \parallel P_G) = \sum_{v \in V} P(\pi_G(v)) \cdot d_{\mathrm{KL}}(P(v \mid \pi_G(v)) \parallel Q(v \mid \pi_G(v))) \geq 0$.

**Corollary 4** (Conditional Mutual Information Tester, adapted from [Bhattacharyya et al., 2021, Theorem 1.3]). *Fix any $\varepsilon > 0$. Let $(X, Y, Z)$ be three random variables over $\Sigma_X, \Sigma_Y, \Sigma_Z$ respectively. Given the empirical distribution $(\hat{X}, \hat{Y}, \hat{Z})$ over a size $N$ sample of $(X, Y, Z)$, there exists a universal constant $0 < C < 1$ so that for any*

$$N \geq \Theta\left(\frac{|\Sigma_X| \cdot |\Sigma_Y| \cdot |\Sigma_Z|}{\varepsilon} \cdot \log \frac{|\Sigma_X| \cdot |\Sigma_Y| \cdot |\Sigma_Z|}{\delta} \cdot \log \frac{|\Sigma_X| \cdot |\Sigma_Y| \cdot |\Sigma_Z| \cdot \log(1/\delta)}{\varepsilon}\right),$$

*the following statements hold with probability $1 - \delta$:*

*(1) If $I(X; Y \mid Z) = 0$, then $\hat{I}(X; Y \mid Z) < C \cdot \varepsilon$.*

*(2) If $\hat{I}(X; Y \mid Z) \geq C \cdot \varepsilon$, then $I(X; Y \mid Z) > 0$.*

*(3) If $\hat{I}(X; Y \mid Z) \leq C \cdot \varepsilon$, then $I(X; Y \mid Z) < \varepsilon$.*

*Unconditional statements involving $I(X; Y)$ and $\hat{I}(X; Y)$ hold similarly by choosing $|\Sigma_Z| = 1$.*

### 2.3 Graph definitions

Let $G = (V, E)$ be a graph on $|V| = n$ vertices and $|E|$ nodes where adjacencies are denoted with dashes, e.g. $u - v$. For any vertex $v \in V$, we use $N(v) \subseteq V \setminus \{v\}$ to denote the neighbors of $v$ and $d(v) = |N(v)|$ to denote $v$'s degree. An undirected cycle is a sequence of $k \geq 3$ vertices such that $v_1 - v_2 - \ldots - v_k - v_1$. For any subset $E' \subseteq E$ of edges, we say that the graph $H = (V, E')$ is the edge-induced subgraph of $G$ with respect to $E'$.

For oriented graphs, we use arrows to denote directed edges, e.g. $u \to v$. We denote $\pi(v)$ to denote the parents of $v$ and $d^{in}(v)$ to denote $v$'s incoming degree. An interesting directed subgraph on three vertices is the v-structure, where $u \to v \leftarrow w$ and $u \nmid w$; we say that $v$ is the center of the v-structure. In this work, we study a generalized higher-degree version of v-structures: we define the notion of *deg-$\ell$ v-structure* as a node $v$ with $\ell \geq 2$ parents $u_1, u_2 \ldots, u_\ell$. We say that a deg-$\ell$ v-structure is said to be $\varepsilon$-strong if we can reliably identify them in the finite sample regime. In our context, it means that for all $k \in [\ell]$, $I(u_k; \{u_1, u_2 \ldots, u_\ell\} \setminus u_k \mid v) \geq C \cdot \varepsilon$, for the universal constant $0 < C < 1$ appearing in Corollary 4. A directed acyclic graph (DAG) is a fully oriented graph without any directed cycles (a sequence of $k \geq 3$ vertices such that $v_1 \to v_2 \to \ldots \to v_k \to v_1$) and are commonly used to represent the conditional dependencies of a Bayes net.

For any partially directed graph, an *acyclic completion* or *consistent extension* refers to an assignment of edge directions to unoriented edges such that the resulting fully directed graph has no directed cycles; we say that a DAG $G$ is *consistent* with a partially directed graph $H$ if $G$ is an acyclic completion of $H$. Meek rules are a set of 4 edge orientation rules that are sound and complete with respect to any given set of arcs that has a consistent DAG extension Meek [1995]. Given any edge orientation information, one can always repeatedly apply Meek rules till a fixed point to maximize the number of oriented arcs. One particular orientation rule (Meek $R1$) orients $b \to c$ whenever a partially oriented graph has the configuration $a \to b - c$ and $a \nmid c$ so as to avoid forming a new v-structure of the form $a \to b \leftarrow c$. In the same spirit, we define Meek $R1(d^*)$ to orient all incident unoriented edges away from $v$ whenever $v$ already has $d^*$ parents in a partially oriented graph.

The *skeleton* $\text{skel}(G)$ of a graph $G$ refers to the resulting undirected graph after unorienting all edges in $G$, e.g. see Fig. 1. A graph $G$ is a *polytree* if $\text{skel}(G)$ is a forest. For $d \geq 1$, a polytree $G$ is a *$d$-polytree* if all vertices in $G$ have at most $d$ parents. Without loss of generality, by picking the minimal $d$, we may assume that $d$-polytrees have a vertex with $d$ parents. When we *freely orient* a forest, we pick arbitrary root nodes in the connected components and orient to form a 1-polytree.

## 3 Recovering a good orientation given a skeleton and degree bound

In this section, we describe and analyze an algorithm for estimating a probability distribution $P$ that is defined on a $d^*$-polytree $G^*$. We assume that we are given $\text{skel}(G^*)$ and $d$ as input.

Note that for some distributions there could be more than one ground truth graph, e.g. when the Markov equivalence class has multiple graphs. In such situations, for analysis purposes, we are free to choose any graph that $P$ is Markov with respect to. As the mutual information scores[3] are the same for any graphs that $P$ is Markov with respect to, the choice of $G^*$ does not matter here.

---

[3]The mutual information score is the sum of the mutual information terms as in Eq. (1).

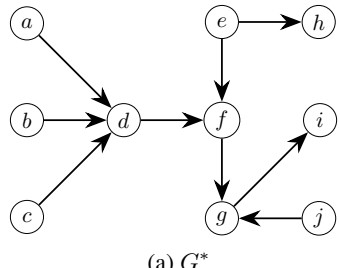 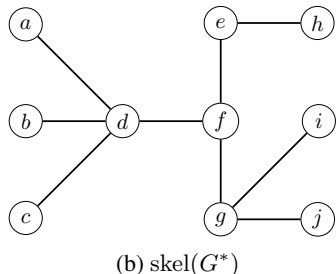

(a) $G^*$          (b) $\mathrm{skel}(G^*)$

Figure 1: Running polytree example with $d^* = 3$ where $I(a; b, c) = I(b; a, c) = I(c; a, b) = 0$ due to the deg-3 v-structure centered at $d$. Since $I(a; f \mid d) = 0$, Corollary 4 tells us that $\hat{I}(a; f \mid d) \leq C \cdot \varepsilon$. Thus, we will *not* detect $a \rightarrow d \rightarrow f$ erroneously as a strong deg-2 v-structure $a \rightarrow d \leftarrow f$.

## 3.1 Algorithm

At any point in the algorithm, let us define the following sets. Let $N(v)$ be the set of all neighbors of $v$ in $\mathrm{skel}(G^*)$. Let $N^{in}(v) \subseteq N(v)$ be the current set of incoming neighbors of $v$. Let $N^{out}(v) \subseteq N(v)$ be the current set of outgoing neighbors of $v$. Let $N^{un}(v) \subseteq N(v)$ be the current set of unoriented neighbors of $v$. That is,

$$N(v) = N^{in}(v) \; \dot{\cup} \; N^{out}(v) \; \dot{\cup} \; N^{un}(v)$$

---

**Algorithm 1** Algorithm for known skeleton and max in-degree.

---

    **Input**: Skeleton $\mathrm{skel}(G^*) = (V, E)$, max in-degree $d^*$, threshold $\varepsilon > 0$, universal constant $C$
    **Output**: A complete orientation of $\mathrm{skel}(G^*)$
1: Run Phase 1: Orient strong v-structures (Algorithm 3)          $\triangleright \; \mathcal{O}(n^{d^*})$ time
2: Run Phase 2: Local search and Meek $R1(d^*)$ (Algorithm 4)      $\triangleright \; \mathcal{O}(n^3)$ time
3: Run Phase 3: Freely orient remaining unoriented edges (Algorithm 5)    $\triangleright \; \mathcal{O}(n)$ time via DFS
4: **return** $\hat{G}$

---

There are three phases to our algorithm. In Phase 1, we orient strong v-structures. In Phase 2, we locally check if an edge is forced to orient one way or another to avoid incurring too much error. In Phase 3, we orient the remaining unoriented edges as a 1-polytree. Since the remaining edges were not forced, we may orient the remaining edges in an arbitrary direction (while not incurring "too much error") as long as the final incoming degrees of any vertex does not increase by more than 1. Subroutine `Orient` (Algorithm 2) performs the necessary updates when we orient $u - v$ to $u \rightarrow v$.

---

**Algorithm 2** `Orient`: Subroutine to orient edges

---

    **Input**: Vertices $u$ and $v$ where $u - v$ is currently unoriented
1: Orient $u - v$ as $u \rightarrow v$.
2: Update $N^{in}(v)$ to $N^{in}(v) \cup \{u\}$ and $N^{un}(v)$ to $N^{un}(v) \setminus \{u\}$.
3: Update $N^{out}(u)$ to $N^{out}(u) \cup \{v\}$ and $N^{un}(u)$ to $N^{un}(u) \setminus \{v\}$.

---

## 3.2 Analysis

Observe that we perform $\mathcal{O}(n^{d^*})$ (conditional) mutual information tests in Algorithm 1. The following lemma (Lemma 5) ensures us that *all* our tests will behave as expected with sufficient samples. As such, in the rest of our analysis, we analyze under the assumption that our tests are correct.

**Lemma 5.** *Suppose all variables in the Bayesian network has alphabet $\Sigma$. For $\varepsilon' > 0$, with*

$$m \in \mathcal{O}\left( \frac{|\Sigma|^{d^*+1}}{\varepsilon'} \cdot \log \frac{|\Sigma|^{d^*+1} \cdot n^{d^*}}{\delta} \cdot \log \frac{|\Sigma|^{d^*+1} \cdot \log(n^{d^*}/\delta)}{\varepsilon'} \right)$$

*empirical samples, $\mathcal{O}(n^{d^*})$ statements of the following forms, where $\mathbf{X}$ and $\mathbf{Y}$ are variable sets of size $|\mathbf{X} \; \dot{\cup} \; \mathbf{Y}| \leq d$ and $Z$ is possibly $\emptyset$, all succeed with probability at least $1 - \delta$:*

189 *(1) If $I(\mathbf{X}; \mathbf{Y} \mid Z) = 0$, then $\hat{I}(\mathbf{X}; \mathbf{Y} \mid Z) < C \cdot \varepsilon'$,*

190 *(2) If $\hat{I}(\mathbf{X}; \mathbf{Y} \mid Z) \geq C \cdot \varepsilon'$, then $I(\mathbf{X}; \mathbf{Y} \mid Z) > 0$,*

191 *(3) If $\hat{I}(\mathbf{X}; \mathbf{Y} \mid Z) \leq C \cdot \varepsilon'$, then $I(\mathbf{X}; \mathbf{Y} \mid Z) < \varepsilon'$.*

192 *Proof.* Use Corollary 4 and apply union bound over $\mathcal{O}(n^d)$ tests. □

193 Recall that $\pi(v)$ is the set of true parents of $v$ in $G^*$. Let $H$ be the forest induced by the remaining
194 unoriented edges after phase 2. Let $\hat{G}$ be returned graph of the algorithm 1. Let us denote the final
195 $N^{in}(v)$ as $\pi^{in}(v)$ at the end of Phase 2, just before freely orienting, i.e. the vertices pointing into $v$
196 in $\hat{G} \setminus H$. Let $\pi^{un}(v) = \pi(v) \setminus \pi^{in}(v)$ be the set of ground truth parents that are not identified in
197 Phase 1. Since the algorithm does not make mistakes for orientations in $\hat{G} \setminus H$ (Lemma 6), all edges
198 of in $\pi^{un}(v)$ will be unoriented.

199 **Lemma 6.** *Any oriented arc in $\hat{G} \setminus H$ is a ground truth orientation. That is, any vertex parent set*
200 *in $\hat{G} \setminus H$ is a subset of $\pi(v)$, i.e. $\pi^{in}(v) \subseteq \pi(v)$, and $N^{in}(v)$ at any time during the algorithm will*
201 *have $N^{in}(v) \subseteq \pi^{in}(v)$.*

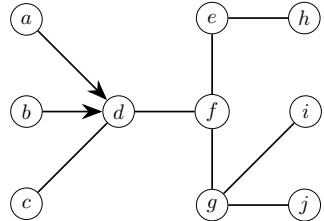

Figure 2: Suppose we have the following partially oriented graph in the execution of Algorithm 4 (after Phase 1). Since $N^{in}(d) = \{a, b\}$, we will check the edge orientations of $c - d$ and $f - d$. Since $I(f; \{a, b\} \mid d) = 0$, we will have $\hat{I}(f; \{a, b\} \mid d) \leq \varepsilon$, so we will *not* erroneously orient $f \to d$. Meanwhile, $I(c; \{a, b\}) = 0$, we will have $\hat{I}(c; \{a, b\}) \leq \varepsilon$, so we will *not* erroneously orient $d \to c$.

202 Let $\hat{\pi}(v)$ be the proposed parents of $v$ output by Algorithm 1. The KL divergence between the true
203 distribution and our output distribution is essentially $\sum_{v \in V} I(v; \pi(v)) - \sum_{v \in V} I(v; \hat{\pi}(v))$ as the
204 structure independent terms will cancel out.

205 To get a bound on the KL divergence, we will upper bound $\sum_{v \in V} I(v; \pi(v))$ and lower bound
206 $\sum_{v \in V} I(v; \hat{\pi}(v))$. To upper bound $I(v; \pi(v))$ in terms of $\pi^{in}(v) \subseteq \pi(v)$ and $I(v; u)$ for $u \in$
207 $\pi^{un}(v)$, we use Lemma 8 which relies on repeated applications of Lemma 7. To lower bound
208 $\sum_{v \in V} I(v; \hat{\pi}(v))$, we use Lemma 9.

209 **Lemma 7.** *Fix any vertex $v$, any $S \subseteq \pi^{un}(v)$, and any $S' \subseteq \pi^{in}(v)$. If $S \neq \emptyset$, then there exists a*
210 *vertex $u \in S \cup S'$ with*

$$I(v; S \cup S') \leq I(v; S \cup S' \setminus \{u\}) + I(v; u) + \varepsilon . \qquad (2)$$

211 **Lemma 8.** *For any vertex $v$ with $\pi^{in}(v)$, we can show that*

$$I(v; \pi(v)) \leq \varepsilon \cdot |\pi(v)| + I(v; \pi^{in}(v)) + \sum_{u \in \pi^{un}(v)} I(v; u) .$$

---

**Algorithm 3** Phase 1: Orient strong v-structures

1: $d \leftarrow d^*$
2: **while** $d \geq 2$ **do**
3:      **for** $v \in V$ **do**                  ▷ Arbitrary order
4:          Let $\mathcal{N}_d \subseteq 2^{N(v)}$ be the set of $d$ neighbors of $v$      ▷ $|\mathcal{N}_d| = \binom{|N(v)|}{d}$
5:          **for** $S \in \mathcal{N}_d$ s.t. $|S| = d$, $|S \cup N^{in}(v)| \leq d^*$, and $\hat{I}(u; S \setminus \{u\} \mid v) \geq C \cdot \varepsilon$, $\forall u \in S$ **do**
6:              **for** $u \in S$ **do**              ▷ Strong deg-$d$ v-structure
7:                  ORIENT($u, v$)
8:      $d \leftarrow d - 1$                  ▷ Decrement degree bound

---

---
**Algorithm 4** Phase 2: Local search and Meek $R1(d^*)$

---
 1: **do**                  ▷ $\mathcal{O}(n)$ iterations, $\mathcal{O}(n^2)$ time per iteration
 2:      **if** $\exists v \in V$ such that $|N^{in}(v)| = d^*$ and $N^{un}(v) \neq \emptyset$ **then**         ▷ Meek $R1(d^*)$
 3:          Orient all unoriented arcs *away* from $v$
 4:          Update $N^{out}(v) \leftarrow N^{out}(v) \cup N^{un}(v)$; $N^{un}(v) \leftarrow \emptyset$
 5:      **for** every node $v \in V$ **do**
 6:          **if** $1 \leq |N^{in}(v)| < d^*$ **then**
 7:              **for** every $u \in N^{un}(v)$ **do**
 8:                  **if** $\hat{I}(u; N^{in}(v) \mid v) > C \cdot \varepsilon$ **then** ORIENT$(u, v)$
 9:                  **else if** $\hat{I}(u; N^{in}(v)) > C \cdot \varepsilon$ **then** ORIENT$(v, u)$
10: **while** new edges are being oriented

---

---
**Algorithm 5** Phase 3: Freely orient remaining unoriented edges

---
 1: Let $H$ be the forest induced by the remaining unoriented edges.
 2: Freely orient $H$ as a 1-polytree (i.e. maximum in-degree in $H$ is 1).
 3: Let $\hat{G}$ be the combination of the oriented $H$ and the previously oriented arcs.
 4: **return** $\hat{G}$

---

In Phase 3, we increase the incoming edges to any vertex by at most one. The following lemma tells us that we lose at most[4] an additive $\varepsilon$ error per vertex.

**Lemma 9.** *Consider an arbitrary vertex $v$ with $\pi^{in}(v)$ at the start of Phase 3. If Phase 3 orients $u \rightarrow v$ for some $u - v \in H$, then*

$$I(v; \pi^{in}(v) \cup \{u\}) \geq I(v; \pi^{in}(v)) + I(v; u) - \varepsilon.$$

By using Lemma 8 and Lemma 9, we can show our desired KL divergence bound (Lemma 10).

**Lemma 10.** *Let $\pi(v)$ be the true parents of $v$. Let $\hat{\pi}(v)$ be the proposed parents of $v$ output by our algorithm. Then,*

$$\sum_{v \in V} I(v; \pi(v)) - \sum_{v \in V} I(v; \hat{\pi}(v)) \leq n \cdot (d^* + 1) \cdot \varepsilon \ .$$

With these results in hand, we are ready to establish our main theorem:

*Proof of Theorem 1.* We first combine Lemma 10 and Lemma 5 with $\varepsilon' = \frac{\varepsilon}{2n \cdot (d^* + 1)}$ in order to obtain an orientation $\hat{G}$ which is close to $G^*$. Now, recall that there exist efficient algorithms for estimating the parameters of a Bayes net with in-degree-$d$ (note that this includes $d$-polytrees) $P$ once a close-enough graph $\hat{G}$ is recovered [Dasgupta, 1997, Bhattacharyya et al., 2020], with sample complexity $\tilde{\mathcal{O}}(|\Sigma|^d n / \varepsilon)$. Denote the final output $\hat{P}_{\hat{G}}$, a distribution that is estimated using the conditional probabilities implied by $\hat{G}$. One can bound the KL divergences as follows:

$$d_{\mathrm{KL}}(P \parallel P_{\hat{G}}) - d_{\mathrm{KL}}(P \parallel P_{G^*}) \leq \varepsilon/2 \quad \text{and} \quad d_{\mathrm{KL}}(P \parallel \hat{P}_{\hat{G}}) - d_{\mathrm{KL}}(P \parallel P_{\hat{G}}) \leq \varepsilon/2 \ .$$

Thus, we see that

$$d_{\mathrm{KL}}(P \parallel \hat{P}_{\hat{G}}) \leq \varepsilon + d_{\mathrm{KL}}(P \parallel P_{G^*}) = \varepsilon \ .$$

$\square$

# 4   Skeleton assumption

In this section, we present a set of *sufficient* assumptions (Assumption 11) under which the Chow-Liu algorithm will recover the true skeleton even while with finite samples.

---

[4]Orienting "freely" could also increase the mutual information score and this is considering the worst case.

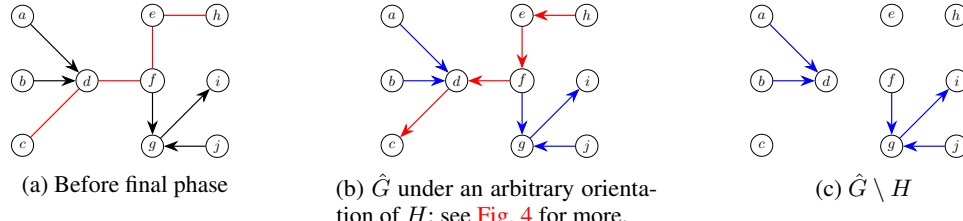

(a) Before final phase

(b) $\hat{G}$ under an arbitrary orientation of $H$; see Fig. 4 for more.

(c) $\hat{G} \setminus H$

Figure 3: Consider the partially oriented graph before the final phase, where $H$ is the edge-induced subgraph on the unoriented edges in red. Since $d^* = 3$ is known, we can conclude that $g \to i$ was oriented due to a local search step and not due to Meek $R1(3)$. We have the following sets before the final phase: $\pi^{in}(c) = \{a, b\}$, $\pi^{in}(g) = \{f, j\}$, $\pi^i = \{g\}$, $\pi^{un}(d) = \{c\}$, $\pi^{un}(f) = \{d, e\}$, and $\pi^{un}(e) = \{h\}$. With respect to the chosen orientation of $H$ and the notation in Lemma 10, we have $A = \{c, d, f, e, h\}$, $a_c = d$, $a_d = f$, $a_f = e$, and $a_e = h$. Observe that the $\pi^{un}$'s and $a$'s are two different ways to refer to the set of red edges of $H$.

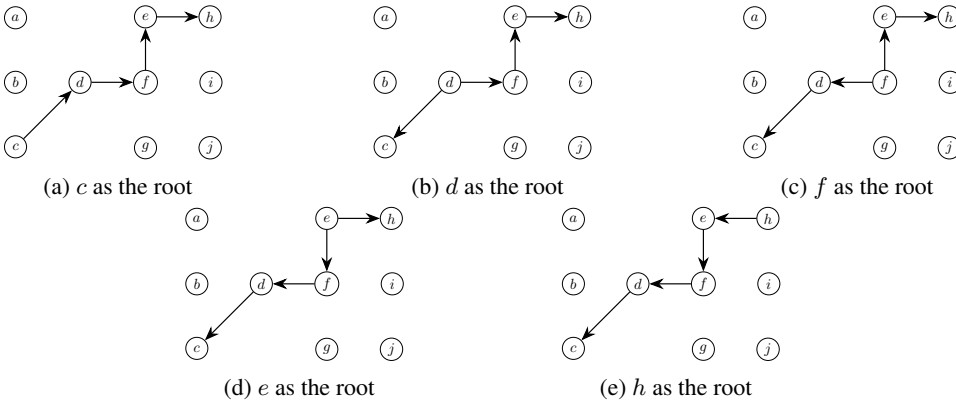

(a) $c$ as the root

(b) $d$ as the root

(c) $f$ as the root

(d) $e$ as the root

(e) $h$ as the root

Figure 4: The five different possible orientations of $H$. Observe that the ground truth orientation of these edges is inconsistent with all five orientations shown here.

**Assumption 11.** For any given distribution $P$, there exists a constant $\varepsilon_P > 0$ such that:
(1) For every pair of nodes $u$ and $v$, if there exists a path $u - \cdots - v$ of length greater than 2 in $G^*$, then then $I(u; v) + 3 \cdot \varepsilon_P \leq I(a; b)$ for every pair of adjacent vertices $a - b$ in the path.
(2) For every pair of directly connected nodes $a - b$ in $G^*$, $I(a; b) \geq 3 \cdot \varepsilon_P$.

Suppose there is a large enough gap of $\varepsilon_P$ between edges in $G^*$ and edges outside of $G^*$. Then, with $\mathcal{O}(1/\varepsilon_P^2)$ samples, each estimated mutual information $\hat{I}(a; b)$ will be sufficiently close to the true mutual information $I(a; b)$. Thus, running the Chow-Liu algorithm (which is essentially maximum spanning tree on the estimated mutual information on each pair of vertices) recovers $\mathrm{skel}(G^*)$.

**Lemma 12.** *Under Assumption 11, running the Chow-Liu algorithm on the $m$-sample empirical estimates $\{\hat{I}(u; v)\}_{u,v \in V}$ recovers a ground truth skeleton with high probability when $m \geq \Omega(\frac{\log n}{\varepsilon_P^2})$.*

Combining Lemma 12 with our algorithm Algorithm 1, one can learn a polytree that is $\varepsilon$-close in KL with $\tilde{\mathcal{O}}\left( \max\left\{ \frac{\log(n)}{\varepsilon_P^2}, \frac{2^d \cdot n}{\varepsilon} \right\} \right)$ samples, where $\varepsilon_P$ depends on the distribution $P$.

## 5 Lower bound

In this section, we show that $\Omega(n/\varepsilon)$ samples are necessary *even when a known skeleton is provided*. For constant in-degree $d$, this shows that our proposed algorithm in Section 3 is sample-optimal up to logarithmic factors.

We first begin by showing a lower bound of $\Omega(1/\varepsilon)$ on a graph with three vertices, even when the skeleton is given. Let $G_1$ be $X \to Z \to Y$ and $G_2$ be $X \to Z \leftarrow Y$, such that $\mathrm{skel}(G_1) = \mathrm{skel}(G_2)$ is $X - Z - Y$. Now define $P_1$ and $P_2$ as follows:

$$P_1 : \begin{cases} X \sim \text{Bern}(1/2) \\ Z = \begin{cases} X & \text{w.p. } 1/2 \\ \text{Bern}(1/2) & \text{w.p. } 1/2 \end{cases} \\ Y = \begin{cases} Z & \text{w.p. } \sqrt{\varepsilon} \\ \text{Bern}(1/2) & \text{w.p. } 1 - \sqrt{\varepsilon} \end{cases} \end{cases} \qquad P_2 : \begin{cases} X \sim \text{Bern}(1/2) \\ Y \sim \text{Bern}(1/2) \\ Z = \begin{cases} X & \text{w.p. } 1/2 \\ Y & \text{w.p. } \sqrt{\varepsilon} \\ \text{Bern}(1/2) & \text{w.p. } 1/2 - \sqrt{\varepsilon} \end{cases} \end{cases} \qquad (3)$$

The intuition is that we keep the edge $X \to Z$ "roughly the same" and tweak the edge $Y - Z$ between the distributions. We define $P_{i,G}$ as projecting $P_i$ onto $G$. One can check that the following holds (see Supplemental for the detailed calculations):

**Lemma 13.** *Let $G_1$ be $X \to Z \to Y$ and $G_2$ be $X \to Z \leftarrow Y$, such that $\text{skel}(G_1) = \text{skel}(G_2)$ is $X - Z - Y$. With respect to Eq. (3), we have the following:*

1. *$d_{\text{H}}^2(P_1, P_2) \in \mathcal{O}(\varepsilon)$*

2. *$d_{\text{KL}}(P_1 \parallel P_{1,G_1}) = 0$ and $d_{\text{KL}}(P_1 \parallel P_{1,G_2}) \in \Omega(\varepsilon)$*

3. *$d_{\text{KL}}(P_2 \parallel P_{2,G_2}) = 0$ and $d_{\text{KL}}(P_2 \parallel P_{2,G_1}) \in \Omega(\varepsilon)$*

Our hardness result (Lemma 14) is obtained by reducing the problem of finding an $\varepsilon$-close graph orientation of $X - Z - Y$ to the problem of *testing* whether the samples are drawn from $P_1$ or $P_2$: To ensure $\varepsilon$-closeness in the graph orientation, one has to correctly determine whether the samples come from $P_1$ or $P_2$ and then pick $G_1$ or $G_2$ respectively. However, it is well-known that distinguishing two distributions whose squared Hellinger distance is $\varepsilon$ requires $\Omega(1/\varepsilon)$ samples (see, e.g., [Bar-Yossef, 2002, Theorem 4.7]).

**Lemma 14.** *Even when given $\text{skel}(G^*)$, it takes $\Omega(1/\varepsilon)$ samples to learn an $\varepsilon$-close graph orientation of $G^*$ for distributions on $\{0, 1\}^3$.*

Using the above construction as a gadget, we can obtain a dependency on $n$ in our lower bound by constructing $n/3$ independent copies of the above gadget, à la proof strategy of Bhattacharyya et al. [2021, Theorem 7.6]. For some constant $c > 0$, we know that a constant $1/c$ fraction of the gadgets will incur an error or more than $\varepsilon/n$ if less than $cn/\varepsilon$ samples are used. The desired result then follows from the tensorization of KL divergence, i.e., $d_{\text{KL}}\left(\prod_i P_i \parallel \prod_i Q_i\right) = \sum_i d_{\text{KL}}(P_i \parallel Q_i)$.

**Theorem 15.** *Even when given $\text{skel}(G^*)$, it takes $\Omega(n/\varepsilon)$ samples to learn an $\varepsilon$-close graph orientation of $G^*$ for distributions on $\{0, 1\}^n$.*

## 6 Conclusion

In this work, we studied the problem of estimating a distribution defined on a $d$-polytree $P$ with graph structure $G^*$ using finite observational samples. We designed and analyzed an efficient algorithm that produces an estimate $\hat{P}$ such that $d_{\text{KL}}(P \parallel \hat{P}) \leq \varepsilon$ assuming access to $\text{skel}(G^*)$ and $d$. The skeleton $\text{skel}(G^*)$ is recoverable under Assumption 11 and we show that there is an inherent hardness in the learning problem even under the assumption that $\text{skel}(G^*)$ is given. For constant $d$, our hardness result shows that our proposed algorithm is sample-optimal up to logarithmic factors.

An interesting open question is whether one can extend the hardness result to arbitrary $d \geq 1$, or design more efficient learning algorithms for $d$-polytrees.

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
