# OpenReview forum: "Learning bounded-degree polytrees with samples"
_NeurIPS.cc/2023/Conference — Submitted to NeurIPS 2023_

### Official Review · Reviewer_TyxN · 2023-06-08

**Soundness:** 3 good
**Presentation:** 2 fair
**Contribution:** 2 fair
**Rating:** 6
**Confidence:** 2

**Summary:**

The paper describes an approach for learning bounded in-degree polytrees that a family of Bayesian networks. More precisely, given the skeleton of the polytree $P$ from which the samples are from, their algorithm learns a $d$-polytree whose distribution is likely to be close to $P$ (with respect to KL divergence) using mutual information tests. Importantly, the algorithm runs in polynomial time for a fixed $d$, whereas the exact learning problem is known to be NP-hard for $d > 1$.

**Strengths:**

To my knowledge, the theoretical results are novel and show that even though the optimal $d$-polytrees are hard to learn exactly, it can be done approximatively.

**Weaknesses:**

My main concern is in the relevance of the article to AI community, i.e., it has nice theoretical results, but their practicality remains unclear to me (see Questions section of this review). I would be happy to increase my score if the authors can offer convincing arguments for this.

I also recommend carefully proofreading the paper to improve its presentation. To mention some of the minor issues:

- 139: "We denote $\pi(v)$ to denote"
- 143: The definition of deg-l v-structure should probably include the lack of edges between $u_i$ and $u_j$? Of course, that holds implicitly for forests.
- 143: "We say that -- is said to be"
- 153: Meek [1995] -> [Meek, 1995]
- 186: has -> have

**Questions:**

What kinds of instances would the described algorithms be practical for? In the description of the Algorithm 3, you iterate over all $O(n^d)$ sets of neighbors of size $d$ and compute the estimated mutual information for them. If additionally the number of samples is of order of magnitude $2^d n / \epsilon$ (line 242), for how large $n$ and $d$ would the described algorithm run in a reasonable time?

In Lemma 5, you state that the mutual information tests succeed with probability at least $1 - \delta$. However, for example, line 197 states that "the algorithm does not make mistakes for orientations" and line 201 seems to imply that $\hat{I}$ must always be less than $\epsilon$. Am I misunderstanding something or shouldn't there be a risk of erroneus orientations?

**Limitations:**

See Questions.

---

> ### Author Rebuttal · Authors · 2023-08-09
>
> **Motivation and practicality**
>
> The structure learning problem for high-dimensional distributions has been widely studied in the machine learning community for the last four decades (e.g., Chapters 16-20 of [KF09]). In particular, learning polytree Bayes nets is of great interest, because polytree models admit efficient exact inference using a classic belief-propagation algorithm [KP83]. The book [PNM08] gives a comprehensive, though a bit outdated, survey of applications of Bayes nets.
>
> The focus of our work is to advance our understanding of the fundamental achievability and limits of learning high-dimensional graphical models. While advancing the performance of structure learning algorithms in practice is very important in its own right, this is something that is out of the scope of our current work. We note that such theoretical results are explicitly in scope for NeurIPS (cf. the call for paper, and the “Theory” topic within).
>
> **Typos**
>
> Thank you for catching the typos (all are valid). We will fix them in our revision and also do a more careful proofreading.
>
> **Confusion about Lines 197 and 201**
>
> Our analysis operate under the event that Lemma 5 succeeds, which happens with probability at least $1-\delta$. Under this event, the orientations made in our algorithm are guaranteed to be correct due to Lemma 5 point 1 and $C < 1$. In the event that any single subroutine fails, we declare that the whole algorithm failed. However, this happens with very low probability (at most $\delta$) and we can reduce the failure probability by supplying more samples.
>
> **References**
>
> [KF09] Koller, Friedman. Probabilistic Graphical Models - Principles and Techniques. MIT Press, 2009.
>
> [PNM08] Pourret, Na, Marcot. Bayesian networks: a practical guide to applications. John Wiley \& Sons, 2008.
>
> [KP83] Kim, Pearl. A computational model for causal and diagnostic reasoning in inference systems. IJCAI 1983.

---

> > ### Comment · Reviewer_TyxN · 2023-08-11
> > **Response to Authors**
> >
> > I thank the authors for their response. Although I still have some concerns about the computational details and expressive capabilities of d-polytrees in the sense of the number of distributions they can represent, I am satisfied with the answers. I have updated my rating accordingly.

---

> > > ### Author Response · Authors · 2023-08-20
> > >
> > > Thank you for increasing the score! In our revision, we will incorporate the points discussed in the rebuttal.

---

### Official Review · Reviewer_tvTb · 2023-07-04

**Soundness:** 4 excellent
**Presentation:** 4 excellent
**Contribution:** 4 excellent
**Rating:** 6
**Confidence:** 4

**Summary:**

This paper considers the number of samples to learn a particular class of distributions: bounded-degree polytrees (Bayesian networks whose skeleton is a forest). Recent work has shown that tree-structured Bayesian networks (1-polytrees) are learnable with finite samples; this work makes progress on the natural generalization to polytrees, showing a positive result when the skeleton is given. The work also provides some conditions under which the skeleton is learnable, and a lower bound for the number of samples required.

**Strengths:**

Learning a distribution approximately from finite samples is one of the most fundamental tasks in learning theory. This study of the finite-sample learnability of polytrees is a very natural step for building our understanding of this problem, particularly in the context of the recent work showing learnability for tree-structured models.

The main result of Theorem 1 (finite-sample learnability of degree-bounded polytrees given the skeleton) is quite fundamental. The algorithm and proof are generally quite natural, and furthermore they help demonstrate the clean manner in which the mutual information tester machinery of [Bhattacharyya et al., 2021] can be leveraged for such results. While accompanying results in Section 4 (Skeleton assumption) and Section 5 (Lower bounds) are less surprising, their presence adds more completeness to the general picture.

The paper is generally well-written.

**Weaknesses:**

More motivation for studying polytrees might be appreciated by the general NeurIPS community. Regardless, Bayesian networks are well-motivated and polytrees are a natural continuation of the aforementioned recent work.

The assumption of being given the skeleton is perhaps the most unsatisfying aspect of these results. For context, my understanding is that when learning tree-structured models (as is the focus of the main prior work of [Bhattacharyya et al., 2021]), the entire task is determining the skeleton, as any rooting of the tree is equivalent. In this sense, it is somewhat disappointing that the entire task of the main prior work needs to be given to the polytree learning algorithm. It would be nice to know whether this assumption is inherently required or just an artifact of the current algorithm.

**Questions:**

Is there more discussion that you could provide regarding the necessity of assuming the skeleton is known? Here are some questions that may help orient the discussion (although I do not necessarily expect these to be reasonable to answer in this scope):
* Generally, is there clear intuition whether a similar result to Theorem 1 should hold without being given the skeleton? On one hand, it seems plausible to imagine that if the skeleton is hard to learn, then perhaps the choice of skeleton is not so important. On the other hand, it seems plausible that it is hard to learn the skeleton and there are many approximately correct skeletons, but orienting an only approximately correct skeleton is hard (maybe this has some connection to the hardness in the unrealizable setting).
* Is it clear that the Chow-Liu algorithm does not learn an approximately correct skeleton? (Of course, whether having such a skeleton is helpful seems not necessarily obvious.)

Minor remarks:
* Should it be $\hat{I}$ in line 145?
* Line 194 “algorithm 1” should be capitalized/linked.
* Line 269 “or” -> “of”
* Generally, it seems like there may be minor errors in comments involving $I, \hat{I}$, and $C$. For example, on lines 422-423 it says “$\hat{I}(\dots) \le \varepsilon$. Since $0<C<1$, this implies that $\hat{I}(\dots)\le C \cdot \varepsilon$”. Since $C<1$ it is not clear to my why such an implication would hold. A similar remark is made on lines 436-437.

**Limitations:**

The limitations are addressed fairly in the paper.

---

> ### Author Rebuttal · Authors · 2023-08-09
>
> **Typos regarding $\hat{I}$ and $C \cdot \varepsilon$**
>
> Thank you for pointing this out, we indeed compare $\hat{I}$ with $C \cdot \varepsilon$. We will fix these in our revision.
>
> **Other typos and writing suggestions under minor remarks**
>
> Thank you. We will fix the typos and incorporate your writing suggestions in our revision.

---

> > ### Comment · Reviewer_tvTb · 2023-08-13
> >
> > I thank the authors for their response, and particularly their personal intuitions regarding key difficulties for the task without assuming knowledge of the skeleton. I still believe the primary weakness is how the paper does not resolve whether such an assumption is truly necessary. I have kept the rating as-is.

---

> > > ### Author Response · Authors · 2023-08-20
> > >
> > > Thank you once again for your review! In our revision, we will incorporate the points discussed in the rebuttal.

---

### Official Review · Reviewer_ZTUL · 2023-07-06

**Soundness:** 3 good
**Presentation:** 3 good
**Contribution:** 3 good
**Rating:** 7
**Confidence:** 3

**Summary:**

This paper introduces an efficient learning algorithm for bounded degree polytrees and establishes finite-sample guarantees. Explicit sample complexity and polynomial time complexity are provided. An information-theoretic lower bound is provided, which shows that the sample complexity of the algorithm is nearly tight.

**Strengths:**

The paper provides a novel algorithm for learning d-polytrees with general d, extending a previous algorithm for d=1. The theoretical analysis shows that the algorithm is nearly tight in terms of sample complexity. The results do not require distributional assumptions such as strong faithfulness. The ideas and results are clearly presented in the paper.

**Weaknesses:**

The recovery of the true skeleton relies on Assumption 11. It would be nice if some comments on this assumption could be given (e.g. whether it is expected to be tight)

**Questions:**

The recovery of the true skeleton relies on Assumption 11. Is this assumption expected to be tight, and what is the obstacle for skeleton recovery in more general scenarios?

In line 233, there seems to be a redundant "then"

**Limitations:**

The authors have adequately addressed the limitations in the paper.

---

> ### Author Rebuttal · Authors · 2023-08-09
>
> **Tightness and violation of Assumption 11**
>
> You are right that the skeletal assumption is crucial in our algorithm and analysis. The sufficient condition is a useful proxy check for the applicability of our methods. If one believes that the sufficient conditions hold in a dataset of interest, then one can be assured that the theoretical guarantees follow. We are unaware whether Assumption 11 is tight. Meanwhile, if Assumption 11 is violated, then the natural algorithm to recover skeleton by running Chow-Liu may fail.
> One example is to consider the ground truth skeleton of $X-Y-Z$ where $\max(I(X, Y), I(Y, Z)) \ll \varepsilon$, and $\varepsilon$ is the accuracy parameter. Due to sampling error (of using only $Poly(1/\varepsilon)$ finite samples), $\hat{I}(X, Z)$ could potentially be larger than $\hat{I}(X, Y) \text{ and } \hat{I}(Y, Z)$; and thus could have $X-Z$ connected as a result of running Chow-Liu.
> Moreover, there may be other ways to recover the true skeleton under other sets of assumptions; e.g, the results of [BH20] apply for more general Bayes nets under different assumptions. Meanwhile, note that the information theoretic lower bound in Section 5 that we give holds even when the skeleton is known.
>
> **Typo on Line 233**
>
> Thank you for pointing out the typo. We will fix it in our revision.
>
> **References**
>
> [BH20] Bank, Honorio. Provable Efficient Skeleton Learning of Encodable Discrete Bayes Nets in Poly-Time and Sample Complexity. ISIT 2020.

---

> > ### Comment · Reviewer_ZTUL · 2023-08-16
> >
> > Thank you for the detailed response!

---

> > > ### Author Response · Authors · 2023-08-20
> > >
> > > Thank you once again for your review! In our revision, we will incorporate the points discussed in the rebuttal.

---

### Official Review · Reviewer_iYmi · 2023-07-06

**Soundness:** 3 good
**Presentation:** 3 good
**Contribution:** 3 good
**Rating:** 6
**Confidence:** 2

**Summary:**

The paper gives an efficient PAC-learning algorithm for learning graphical models called "bounded polytrees". These are distributions where 1) the undirected skeleton of the graph is a forest and 2) the in-degree of every node is bounded by some constant $d$. This extends a recent result [1] for directed trees, which is corresponding to the case $d=1$.

In contrast to [1], the paper gives a learning algorithm assuming that the skeleton is given. To achieve that, the estimator of conditional mutual information from [1] is extensively used. This estimator is used in a sequence of clever greedy-like checks in order to orient as many edges as possible. After orienting the remaining edges, it is shown that the resulting distribution must have small KL divergence to the true distribution.

A sufficient condition is also given, under which the skeleton can be learned for certain distributions by the Chow-Liu algorithm (so it does not have to be given to the algorithm). Finally, a lower bound on sample complexity is proved, roughly matching the upper bound of the algorithm in the case of binary alphabet.

[1] Bhattacharyya, Gayen, Price, Vinodchandran, "Near-optimal learning of tree-structured distributions by Chow-Liu", STOC 2021.

**Strengths:**

* The studied problem of efficient learning of graphical models is important and interesting.

* The paper considers distributions with a tree skeleton and arbitrary orientation of edges as opposed to just directed trees. This is a natural and long-studied class of distributions.

* Even given the estimator from [1], the algorithm and proofs are interesting and not trivial.

* Section 3 gives a good outline of the algorithm and its correctness proof and the figures were helpful to me.

**Weaknesses:**

* The algorithm requires the skeleton as input, which I think is a significant limitation. It is not clear how useful is the sufficient condition proposed by the authors in order to remove this limitation.

* The writing could be clearer. Especially the steps which I assume are more standard/obvious to the authors felt rushed. In my opinion, a few places could be rewritten in order to be clearer and more self-contained.

**Questions:**

* The proof of Theorem 1 in lines 219-227 is very fast. You state the part about estimating the conditional distributions in a conclusory way. I thought the reference for this would be Theorem 1.4 in [1], but you cite two other papers instead. And the sample complexity there seems to have factor $|\Sigma|^{d+1}$, not $|\Sigma|^d$. Sorry if I am misunderstanding.
The formulas in lines 225-227 are given without any justification and connecting them to the discussion before.

* In the proof of Lemma 7, lines 421-422, I cannot see why Phase 1 guarantees the existence of this vertex. Can you explain? (By the way, the proof says $u\in S$ and the statement $u\in S\cup S'$. Which is it?) (Also I don't understand why it says $\hat{I}<\epsilon$ if the algorithm is always checking against $C\epsilon$.)

* Section 4 also moves fast. In Assumption 11, do you mean that given $P$ and $G^*$, the assumption holds for those $P$ and $G^*$ (and then $G^*$ will be recovered by Chow-Liu run on $P$)? This is not clear from the writing. Also, is $P$ any distribution or is it coming from a tree?

* In section 5, Lemma 14 seems given without proof as a direct consequence of Lemma 13. Shouldn't you also exclude the possibility that there exists a distribution $X\leftarrow Z\rightarrow Y$ which is close both to $P_1$ and $P_2$?
Also the argument for Theorem 15 is sketchy. I would appreciate a proof in the appendix.

* There seem to be some basic properties that you keep applying without ever mentioning them. For example, if $v$ is a node with in-neighbors $u_1,\ldots,u_k$ then $u_1,\ldots,u_k$ are independent. And then there is a similar fact with out-neighbors and conditional independence. It would be nice to state those facts at least once in the preliminaries.

* Similarly, it would have helped me if you reminded me from time to time when you are applying formula (1).

* lines 202-204: Am I understanding correctly that the formula you are giving here is the KL divergence between $P=P_{G^*}$ and $P_{\hat{G}}$? If yes, why not write it explicitly? Why are you using the word "essentially" which suggests to me that the formula is not entirely correct? Is there a problem I am not seeing?



minor:
* I think sometimes you use $d$ as the true maximum in-degree and sometimes as a variable (e.g., Algorithm 3, proof of Lemma 6). I would avoid this.

* Similarly, the way you use $\epsilon$ and $\epsilon'$ is confusing. The value of $\epsilon$ in Lemmas 7-10 is what you call $\epsilon'$ in the proof of Theorem 1, correct? If yes, maybe you could give the formula for $\epsilon'$ before you start analyzing the algorithm.

* I did not understand much from lines 111-116. In the first sentence, do you mean we can always obtain such a graph for any distribution? I guess you can always take the complete graph, but probably that's not what you mean?

* In line 183, can you justify the $O(n^d)$ bound?

* line 192, $d$ should be $d^*$?

* Line 196-197, do you mean not identified in phase 1, or phases 1 and 2?

* Proof of Lemma 21, second and third line after 491, $I(X;Y)$ should be $I(Z;Y)$?

* typos line 59 "are", line 94 "denotes", line 186 "has", line 198 "in", double-check the notation in the caption of Figure 3,

**Limitations:**

see above

---

> ### Author Rebuttal · Authors · 2023-08-09
>
> **Citation in proof of Theorem 1**
> We agree that a reference for Theorem 1.4 in [BGPV21] should be given here as it is the same proof idea: given a good enough graph, one can apply the parameter learning algorithms referenced in our submission; see line 223.
>
> **Lines 225-227**
> We will add the following elaboration for the inequalities in our paper revision: The first inequality comes from our graph learning algorithm's guarantees while the second inequality comes from performing parameter learning algorithms on the learnt graph referenced on Line 223. The final inequality is implied by these two inequalities.
>
> **Proof of Lemma 7, Lines 421-422**
> You are right, we could just write $u \in S$ instead of $u \in S \cup S'$. We will fix in this in our revision. Thank you for pointing this out.
>
> **Typos regarding $\hat{I}$ and $C \cdot \epsilon$**
> Thank you for pointing this out, we indeed compare $\hat{I}$ with $C \cdot \epsilon$. We will fix these in our revision.
>
> **Assumption 11**
> Assumption 11 is well-defined for any distribution $P$, regardless of whether it is a polytree or what its underlying graph $G^*$ looks like. In the event that $P$ is a polytree, Lemma 12 tells us that Chow-Liu will return the true skeleton $G^*$ whenever Assumption 11 holds for $P$. Note that when we *know* that $P$ is a tree, then Assumption 11 is not even required in the context of PAC learning; see [BGPV21].
>
> **Lemma 14**
> Lemma 14 is indeed a direct consequence of Lemma 13 via the following contrapositive implication: If one can solve the problem in Lemma 14, then one can use that algorithm to solve the problem in Lemma 13. We will add this clarification in our revision. Regarding your other concern, observe that the orientations $\{X \gets Z \to Y, X \gets Z \gets Y, X \to Z \to Y\}$ are equivalent with respect to Equation (1) and we have analyzed $X \to Z \to Y$ in Lemma 13; in fact, these three orientations belong in the same Markov equivalence class.
>
> **Theorem 15**
> We will add a the following paragraph in the appendix.
> Consider a distribution $P$ on $n/3$ independent copies of the lower bound construction from Lemma 14, where each copy is indexed by $P_i$ for $i \in \{1, \ldots, n/3\}$.
> Suppose, for a contradiction, that the algorithm draws $c n/\epsilon$ samples for sufficiently small $c > 0$, and manages to output $Q$ that is $\epsilon$-close to $P$ with probability at least 2/3.
> From Lemma 14 with error tolerance $\Omega(\epsilon / n)$, we know that each copy is *not* $\Omega(\epsilon / n)$-close with probability at least 1/5.
> By Chernoff bound, at least $\Omega(n)$ copies are *not* $\Omega(\epsilon / n)$-close with probability at least 2/3.
> Then, by the tensorization of KL divergence, we see that $d_{\text{KL}} \left( \prod_{i=1}^{n/3} P_i || \prod_{i=1}^{n/3} Q_i \right) = \sum_{i=1}^{n/3} d_{\text{KL}}(P_i ||  Q_i) > \Omega(\epsilon)$.
> This is a contradiction to the assumption that the algorithm produces $Q$ that is $\epsilon$-close to $P$ with probability at least 2/3.
>
> **Basic properties about polytrees and referencing**
> Thank you for your suggestion. We will include these common properties for polytrees into the preliminaries and try to reference them whenever we invoke these properties, including formula (1).
>
> **Lines 202-204**
> Your understanding is correct. We will write this more explicitly in the revision and not use the term ``essentially''.
>
> **$d$ as maximum in-degree and variable**
> Thank you for your suggestion. We will make the notation less confusing in our notation. To be precise, we will keep $d^*$ as the maximum in-degree and replace the notation for degree variable $d$ by $\gamma$.
>
> **$\epsilon$ and $\epsilon'$**
> Yes, your understanding is correct. Thank you for your suggestion, we will shift the definition of $\epsilon'=\frac{\epsilon}{2 n \cdot (d^* + 1)}$ upwards from its current position on Line 220 to appear around the time we state Lemma 5, with appropriate signposting.
>
> **Discussion on Lines 111-116**
> You are right that one can always pick $G$ to be a clique in order to satisfy the $\epsilon$-close requirement. However, we are interested in obtaining a graph $G$ that is both $\epsilon$-close and facilitates efficient learning algorithms.
> For instance, if $P$ was defined on a Bayes net with max in-degree $d$, then we want $G$ to also be a Bayes net with max in-degree $d$.
> This is always possible in $\exp{(n)}$ time by formulating the search of $G$ as an optimization problem that maximizes the summation of mutual information term (first term in (1)); see [H93] and why it is NP-hard in general.
> As each mutual information terms can be well-estimated, an $\epsilon$-close graph could be obtained by optimizing over the empirical mutual information scores.
> We will add a version of this discussion in our revision.
>
> **The bound on Line 183**
> Thank you for catching the mistake, it should be $\mathcal{O}(n^{d^*+1})$:
> Since $\binom{n}{k} = \frac{n!}{k! (n-k)!} \leq \frac{n^k}{k!}$, we see that $n \cdot \sum_{k = 1}^{d^\*} \binom{n}{k} \leq n \cdot \sum_{k = 1}^{d^\*} \frac{n^k}{k!} \leq n \cdot n^{d^\*} \sum_{k = 1}^{d^\*} \frac{1}{k!} \leq n \cdot n^{d^\*} \cdot e$.
> That is, we get a bound of $\mathcal{O}(n^{d^\*+1})$.
>
> **Typo on Line 192**
> Thank you, we will correct this.
>
> **Lines 196-197**
> Thank you, we indeed mean ``not identified in both phase 1 and phase 2''. We will correct it in the revision.
>
> **Proof of Lemma 21**
> Thank you for pointing that out. The terms $I(X;Y)$ should indeed by $I(Z;Y)$. We will fix this in the revision.
>
> **typos line 59 "are", line 94 "denotes", line 186 "has", line 198 "in", double-check the notation in the caption of Figure 3**
> Thank you very much for pointing out these mistakes. We will fix them in the revision.
>
> **References**
> [BGPV21] Bhattacharyya, Gayen, Price, Vinodchandran. Near-optimal learning of tree-structured distributions by Chow-Liu. STOC 2021.
> [H93] Höffgen. Learning and robust learning of product distributions. COLT 1993.

---

> > ### Comment · Reviewer_iYmi · 2023-08-11
> >
> > Thank you for your patient replies to my questions. I also appreciate the explanations about the true skeleton in the main rebuttal.
> >
> > **Proof of Lemma 7** This is the main remaining place where I would appreciate some clarifications.
> > 1) You say "Phase 1 guarantees that there exists a vertex $u\in S$ such that $\hat{I}(...)\le\varepsilon$". I think I see the idea but I don't think I understand it completely. For example, as written I think it is possible that $|S\cup S'|>d^*$. In that case you shouldn't have even computed the $\hat{I}$ during Algorithm 3, or am I confused?
> > 2) What is the importance of iterating in the decreasing set size order in the main loop of Algorithm 3?
> > 3) In line 421 you claim $I(u;S\cup S'\setminus\{u\})=0$. I don't understand why this is true. (But you don't need it anyway, since you are upper bounding, right?)
> > 4) In line 423 you say "Corollary 4 tells us $I(u;S\cup S'\setminus\{u\})<\varepsilon$". Is this a typo and what you mean here is the conditional mutual information?
> >
> > **Statement of Assumption 11**
> > 1) When you say "path ... of length greater than 2", I think you mean that the length is 2 or more, correct? It would be good to clarify this one way or the other.
> > 2) Is there some significance of $3$ in $3\cdot\varepsilon_P$? Can it be replaced with $\varepsilon_P$?

---

> > > ### Author Response · Authors · 2023-08-14
> > >
> > > **Proof of Lemma 7**
> > >
> > > 1. We apologize for the confusion. One point of confusion could be the dual use of the notation "$S$" in both the algorithm and the lemma. For clarity, we will replace $S$ with $T$ in the algorithm in this response.
> > >
> > > In Lemma 7, $S$ and $S'$ are defined as $S \subseteq \pi^{un}(v)$ and $S' \subseteq \pi^{in}(v)$, so we are always guaranteed that $|S \cup S'| = |\pi(v)| \leq d^*$. Indeed, our previous rebuttal on the proof of Lemma 7 was not quite accurate, and we  would like to further clarify this part. It should be $u \in S \cup S'$ and Phase 1 guarantees that $u \in S \cup S'$ (rather than just $S$).
> > >
> > > To see why, we need to look at line 5 of Algorithm 3  where we check all subsets $T$ of $\pi(v)$ (as well as some other sets) to see if *every* $u \in T$ satisfies $\hat{I}(u; T \backslash \{u\} | v) \geq C \cdot \varepsilon$. From here, we can see that if a subset $T$ of $\pi(v)$ is not *all* oriented into $v$, then we know that from Algo 3, line 5 that there exists some $u \in T$ such that $\hat{I}(u; T \backslash \{u\} | v) < C \cdot \varepsilon$. Applying this to $T = S \cup S'$, where $S$, the set of unoriented neighboring nodes, is not empty, we have our claim.
> > >
> > > We will expand on this argument in the proof of Lemma 7 for the revision.
> > >
> > > 2. Regarding the question about considering subsets in decreasing size: the actual order does not matter, as long as the algorithm considers all possible subsets of neighbors. Here we chose, for convenience and to keep track more easily, to consider them in decreasing size.
> > >
> > > 3. Line 421: As $u$ is a parent of $v$, it is independent of all other parents $\pi(v)$ of $v$. Since $S \cup S' \setminus \{u\} \subseteq \pi(v)$, we also have that $u$ is independent with those variables, i.e. $I(u; S \cup S' \setminus \{u\}) = 0$.
> > >
> > > 4. Line 423: Yes, it should be $I(u; S \cup S' \setminus \{u\} \mid v) \leq \varepsilon$. Thanks for noticing this!
> > >
> > > **Assumption 11**
> > >
> > > Indeed, we mean to consider u-v paths involving at least one additional vertex, say u-w-v. The "3" is an artifact of the proof and you are right that we could have absorbed that constant into the $\varepsilon_P$ term in the statement of Assumption 11. We will update this in our revision to make it cleaner.

---

> > > > ### Comment · Reviewer_iYmi · 2023-08-14
> > > >
> > > > Thank you for the explanations. It seems most of my problems with understanding Lemma 7 were due to the fact that I mixed up the definitions of $N(v)$ and $\pi(v)$.
> > > >
> > > > I thank the authors for their constructive engagement and encourage them to implement various promised clarifications and fixes. At the moment I have increased the paper's score by one.

---

> > > > > ### Author Response · Authors · 2023-08-20
> > > > >
> > > > > Thank you for increasing the score! In our revision, we will incorporate the clarifications and fixes that we have discussed in the rebuttal.

---

### Author Rebuttal · Authors · 2023-08-09

We thank the reviewer for their time and for providing valuable feedback on our paper. In this global response, we address one of the issues raised by multiple reviewers about the known skeleton assumption, sufficiency of the Assumption 11, and the intuition behind the skeleton-based approach.

---

The skeletal assumption is crucial in our algorithm and analysis.

Assumption 11 is a useful proxy check for the applicability of our methods. If one believes that the sufficient conditions hold in a dataset of interest, then one can be assured that our theoretical guarantees follow. We are unaware whether Assumption 11 is tight. There may be other ways to recover the true skeleton under other sets of assumptions.

Knowing the true skeleton is crucial in our analysis as we need a way to compare the output of our algorithm against the ground truth. This is because the KL divergence of two distributions on a polytree is related to the parent sets; see Equation (1).
We found it difficult to design efficient algorithms with provable guarantees without access to the true skeleton.
In Section 5, we give information-theoretic lower bounds under the assumption that the true skeleton is known, showing that the problem is non-trivial even under the known skeleton assumption.

It is natural to ask whether what we can do with access to a false skeleton that is approximately correct (i.e. has some orientation close in KL to the ground truth) produced by running the Chow-Liu algorithm on the sample statistics.
However, it is unclear to us why we can hope to design efficient algorithms with provable guarantees for two reasons:

- The Chow-Liu algorithm only uses order-1 mutual information while the KL divergence of Equation (1) requires information from order-$d$ mutual information. It is unclear why one can hope that this false skeleton would yield provable guarantees with respect to Equation (1).

- An ``approximately correct'' skeleton may have potentially unknown number of edges in the skeleton being wrong and we do not see how to design efficient global orientation algorithms using only statistics from the ground truth samples.

Without the true skeleton, a "local algorithm" (such as ours) can be tricked into some "local optima" and it is hard to argue why the output would obtain "global guarantees" with respect to the parent sets of Equation (1).

---

### Decision · Program_Chairs · 2023-09-21

**Decision:**

Reject

**Comment:**

While the paper is theoretically sound and the problem seems to be non-trivial, Assumption 11 (known skeleton) makes the paper not very useful or well motivated for a ML audience. Indeed, three reviewers raised the issue of the known skeleton. During the rebuttal, the authors agreed that the skeleton assumption is crucial for their result from a technical viewpoint. Still, the motivation of the known skeleton assumption being reasonable or useful in practice is highly unclear.



The title mentioning "learning polytrees" is a bit misleading, as it might seem the paper does not require the skeleton. Although everything is clarified since the abstract, still a better title seems necessary.



Finally, there is a paper that might be useful for the authors: Pieter Abbeel, Daphne Koller, Andrew Y. Ng "Learning Factor Graphs in Polynomial Time and Sample Complexity", JMLR, 2006.

- Parameter learning: Theorem 5 (computational complexity), Theorem 6 (sample complexity), Theorem 9 (Bayes nets)

- Structure learning: Theorem 14 (computational complexity), Theorem 15 (sample complexity)